# Allopurinol attenuates development of *Porphyromonas gingivalis* LPS-induced cardiomyopathy in mice

**Akinaka Morii**[1,2☉], **Ichiro Matsuo**[2,3☉], **Kenji Suita**[1], **Yoshiki Ohnuki**[1], **Misao Ishikawa**[4], **Aiko Ito**[5], **Go Miyamoto**[1,5], **Mariko Abe**[1,5], **Takao Mitsubayashi**[1,5], **Yasumasa Mototani**[1], **Megumi Nariyama**[6], **Ren Matsubara**[1,6], **Yoshio Hayakawa**[7], **Yasuharu Amitani**[8], **Kazuhiro Gomi**[2], **Takatoshi Nagano**[2], **Satoshi Okumura**[1]*

1 Department of Physiology, Tsurumi University School of Dental Medicine, Yokohama, Japan,
2 Department of Periodontology, Tsurumi University School of Dental Medicine, Yokohama, Japan,
3 Department of Oral and Maxillofacial Surgery, Ibaraki Medical Center Tokyo Medical University, Ibaraki, Japan, 4 Department of Oral Anatomy, Tsurumi University School of Dental Medicine, Yokohama, Japan, 5 Department of Orthodontology, Tsurumi University School of Dental Medicine, Yokohama, Japan, 6 Department of Pediatric Dentistry, Tsurumi University School of Dental Medicine, Yokohama, Japan, 7 Department of Dental Anesthesiology, Tsurumi University School of Dental Medicine, Yokohama, Japan, 8 Department of Mathematics, Tsurumi University School of Dental Medicine, Yokohama, Japan

☉ These authors contributed equally to this work.
* okumura-s@tsurumi-u.ac.jp

## Abstract

Oxidative stress is involved in the progression of periodontitis, independently of confounding factors such as smoking, and numerous studies suggest that periodontitis is associated with increased risk of cardiovascular disease. In this study, therefore, we examined the effects of the xanthine oxidase inhibitor allopurinol on cardiac dysfunction in mice treated with *Porphyromonas gingivalis* lipopolysaccharide (PG-LPS) at a dose (0.8 mg/kg/day) equivalent to the circulating level in patients with periodontal disease. Mice were divided into four groups: 1) control, 2) PG-LPS, 3) allopurinol, and 4) PG-LPS + allopurinol. After 1 week, we evaluated cardiac function by echocardiography. The left ventricular ejection fraction was significantly decreased in PG-LPS-treated mice compared to the control (from 68 ± 1.3 to 60 ± 2.7%), while allopurinol ameliorated the dysfunction (67 ± 1.1%). The area of cardiac fibrosis was significantly increased (approximately 3.6-fold) and the number of apoptotic myocytes was significantly increased (approximately 7.7-fold) in the heart of the PG-LPS-treated group versus the control, and these changes were suppressed by allopurinol. The impairment of cardiac function in PG-LPS-treated mice was associated with increased production of reactive oxygen species by xanthine oxidase and NADPH oxidase 4, leading to calmodulin kinase II activation with increased ryanodine receptor 2 phosphorylation. These changes were also suppressed by allopurinol. Our results suggest that oxidative stress plays an important role in the PG-LPS-promoted development of cardiac diseases, and further indicate that allopurinol ameliorates *Porphyromonas gingivalis* LPS-induced cardiac dysfunction.

**Data availability statement:** The authors confirm that all data supporting the finding of this study are available within the article and its Supplementary materials.

**Funding:** This study was supported by the Japan Society for the Promotion of Science (JSPS) KAKENHI Grant (24K20123 to Ichiro Matsuo, 23K09493 to Kenji Suita 23K09517 to Yoshiki Ohnuki, 24K20067 to Aiko Ito, 22K10255 to Megumi Nariyama, and 24K13250 to Satoshi Okumura). The founders had no role in study design, data collection and analysis, decision to publish, or preparation of the manuscript.

**Competing interests:** The authors have declared that no competing interests exist.

## Introduction

Periodontitis is a chronic inflammatory destructive disease in the tooth-supporting tissue—the periodontium. It is a bacterial disease, but the eventual tissue destruction results from the interplay between pathogen activity and host response. Periodontitis is generally accepted as a risk factor for cardiovascular disease (CVD), though the causality still remains debatable [1]. The progression of periodontitis involves oxidative stress due to reactive oxygen species (ROS) produced as a result of disturbance in the regulation of the host inflammatory response to bacterial infection [2]. Oxidative stress affects various organs of the body [3], and so ROS associated with periodontitis might also be related to other systemic diseases, including CVD [3]. Therefore, a proper understanding of oxidative stress and its pathways, including the formation of free radicals and inflammatory markers related to oral diseases, is important for effective treatment. In this context, xanthine oxidase (XO), a purine-catabolic enzyme which is upregulated in the left ventricular tissue in chronic heart failure [4], is of interest, as it generates ROS via several pathways, including calcium signaling [5, 6].

Intracellular $Ca^{2+}$ handling is a critical regulator of action potential duration, as well as the mechanical activity of cardiac myocytes via excitation-contraction coupling [7]. Abnormalities in its homeostasis can therefore reduce cardiac output, which may be fatal, and can also initiate cardiac remodeling and heart failure. Previous studies have reported that XO inhibition has beneficial effects on cardiac remodeling [8–10], mechano-energetics [11] and endothelial function [12] in both experimental and clinical contexts. However, the effect of XO inhibition on CVD in patients with periodontal disease remains unclear.

Therefore, in this study we investigated the potential beneficial effects of allopurinol, a XO inhibitor, on *Porphyromonas gingivalis* LPS-induced cardiac dysfunction in mice.

## Materials and methods

Male 12-week-old C57BL/6 mice obtained from CLEA Japan (Tokyo, Japan) were used in all animal experiments. Mice were group-housed at 23 °C under a 12–12 light/dark cycle with lights on at 8:00 AM in accordance with the standard conditions for mouse studies by our group [13–15]. Both food and water were available ad libitum.

As in our earlier studies [16–18], PG-LPS (#14966–71; Invitrogen, San Diego, CA, USA) was dissolved in saline to prepare a 0.6 mg/ml stock solution, and an appropriate volume of this solution to provide the desired dose (PG-LPS: 0.8mg/kg) was added to 0.2 mL of saline to prepare the solution for intraperitoneal (i.p.) injection (once daily for 1 week). Mice were group-housed (approximately 3 per cage) and were divided into four groups: a normal control group, (Control), a PG-LPS treatment group (PG-LPS), an allopurinol-only treatment group, and a PG-LPS plus allopurinol treatment group (Fig 1). Allopurinol (#A8003; Sigma-Aldrich, St. Louis, MO, USA) was directly dissolved in drinking water (50mg/kg/day; freshly prepared every day) [19, 20]. This model is not a sepsis model, and indeed, no mortality was observed, as expected, because the dose of PG-LPS used in this study is consistent with the circulating levels in patients with periodontitis [21]. After the completion of each treatment, mice were killed by cervical dislocation under anesthesia via a mask with isoflurane (1.0–1.5% v/v) [22]. The heart, lung and liver were excised, weighed, frozen in liquid nitrogen, and stored at −80°C. The ratio of organ mass (mg) to tibial length (TL; mm) was used as an index of organ volume.

### Ethical approval

All animal experiments were complied with the Arrival guidelines [23] and were carried out in accordance with the National Institutes of Health guide for the care and use of laboratory

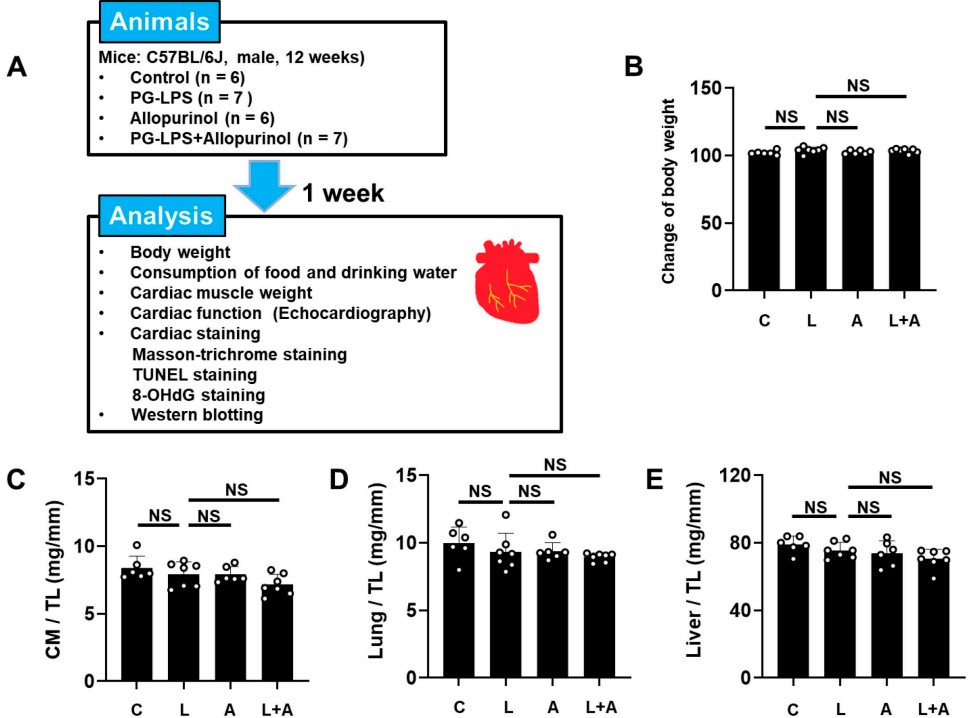

**Fig 1. Schematic illustration of experimental procedure and comparison of body weight, cardiac muscle weight, lung weight and liver weight in the four groups. (A**-B) Control (C), PG-LPS (L), allopurinol (A) and PG-LPS+allopurinol (L + A) groups all showed similar body weight at 1 week after the PG-LPS infusion. NS, not significantly diiferent from the Control ($P > 0.05$) by ANOVA/Tukey-Kramer. (C-E) Cardiac muscle (CM) weight per tibia length (TL) ratio (C), lung weight per TL ratio (D), and liver weight per TL ratio (**E**) were similar among the Control (C), PG-LPS (P), allopurinol (A) and PG-LPS + allopurinol (L + A) groups. all showed similar body weight at 1 week after the PG-LPS infusion. NS, not significantly different from the Control ($P > 0.05$) by non-parametric ANOVA/Steel-Dwass (**C**) or ANOVA/Tukey-Kramer (**D** and **E**).

animals [24] and institutional guidelines. The experimental protocol was approved by the Animal Care and Use Committee of Tsurumi University (No. 29A041).

## Physiological experiments

Echocardiographic measurements were performed at room temperature by means of ultrasonography (TUS-A300, Toshiba, Tokyo, Japan) under anesthesia with isoflurane vapor (1.0–1.5% v/v) titrated to maintain the lightest anesthesia possible [25].

## Evaluation of fibrosis

Cross sections (10 µm) were cut with a cryostat (CM1900; Leica Microsystems, Nussloch, Germany). The sections were air-dried and fixed with 4% paraformaldehyde (v/v) in 0.1 M phosphate-buffered saline (pH 7.5).

Interstitial fibrosis was evaluated by Masson-trichrome staining using the Accustatin Trichrome Stain Kit (#HT15–1KT; Sigma-Aldrich, St. Louis, MO, USA) [13] in accordance with the manufacturer's protocol, as described previously [13,26,27]. We quantified interstitial fibrotic regions from the four groups (Control: $n = 6$, PG-LPS: $n = 6$, allopurinol: $n = 6$, PG-LPS + allopurinol: $n = 7$) using freely available image analysis software (Image J 1.48) to evaluate the percentage of blue area in the Masson-trichrome-stained sections [13,26,27].

## Evaluation of apoptosis

Apoptosis was determined by means of terminal deoxyribonucleotidyl transferase (TdT)-mediated biotin-16-deoxyuridine (TUNEL) staining using an Apoptosis *in situ* Detection kit (#293–71500; Wako, Osaka, Japan). TUNEL-positive nuclei per field of view were manually counted in six sections from the four groups (Control: $n = 5$, PG-LPS: $n = 6$, allopurinol: $n = 6$, PG-LPS + allopurinol: $n = 6$) over a microscopic field of 20 x, averaged and expressed as the ratio of TUNEL-positive nuclei (%) [25,28]. Limiting the counting of total nuclei and TUNEL-positive nuclei to areas with true cross sections of myocytes made it possible to selectively count only those nuclei that were clearly located within myocytes.

## Immunostaining

Oxidative DNA damage in the myocardium was evaluated by immunostaining for 8-hydroxy-2'-deoxyguanosine (8-OHdG) using the Vector M.O.M. Immunodetection system (Control: $n = 5$, PG-LPS: $n = 6$, allopurinol: $n = 6$, PG-LPS + allopurinol: $n = 6$) (#PK-2200, Vector Laboratories, Inc. Burlingame, CA, USA) under our standard conditions [13,26]. Cross sections were cut at 10 μm with a cryostat at –20°C, air-dried and fixed with 4% paraformaldehyde (v/v) in TBS-T for 5 min at room temperature. Antigen retrieval was achieved with 0.1% citrate plus 1% Triton X-100 for 30 min at room temperature, then the sections were washed with TBS-T, incubated with 0.3% horse serum in TBS-T for 1 h at room temperature, and blocked with M.O.M. blocking reagent (Vector Laboratories, Burlingame, CA, USA) overnight at 4°C. For the positive control, sections were incubated with 0.3% $H_2O_2$ in TBS-T before anti-8-OHdG antibody treatment. The sections were incubated with anti-8-OHdG antibody (8.3 μg/ml in M.O.M. Dilute; clone N45.1 monoclonal antibody; Japan Institute for the Control of Aging, Shizuoka, Japan) overnight at 4°C in a humidified chamber, then incubated with 0.3% $H_2O_2$ in 0.3% horse serum for 1 h at room temperature to inactivate endogenous peroxidase, rinsed with TBS-T, incubated with anti-mouse IgG in M.O.M. Diluent, and processed with an ABC kit (Vector Laboratories, Inc. Burlingame, CA, USA). The ratio of 8-OHdG nuclei with oxidative DNA damage (stained dark brown) per total cell number was evaluated.

## Western blotting

Cardiac tissue excised from the mice was homogenized in a Polytron (Kinematica AG, Lucerne, Switzerland) in ice-cold-RIPA buffer (#89900, Thermo Fisher Scientific, Waltham, MA, USA: 25 mM Tris-HCl (pH 7.6), 150 mM NaCl, 1% sodium deoxycholate, 0.1% SDS) with addition of Halt™ Protease Inhibitor Cocktail, EDTA-free (#87785; Thermo Fisher Scientific) and the homogenate was centrifuged at 13,000 x *g* for 10 min at 4°C. The supernatant was collected and the protein concentration was measured using a DC protein assay kit (Bio-Rad, Hercules, CA, USA). Equal amounts of protein (5 μg) were subjected to 12.5% SDS-polyacrylamide gel electrophoresis and blotted onto PVDF membrane (#IPVH00010; Millipore, Burlington, MA, USA).

Western blotting was conducted with commercially available antibodies [25,28–30]. The primary antibodies against collagen 1 (1:1000, AB765P) and oxidized calmodulin kinase II (CaMKII; Met-281/282) (1:1000, 071387) were purchased from Merck (San Jose, CA, USA). Primary antibody against collagen type 3 (1:1000, NB600–594) were purchased from Novus Biological (Centennicas, CO, USA). Primary antibodies against NOX4 (1:1000, #ab 133303) and XO (1:1000, #ab109235) were purchased from Abcam (Cambridge, UK). CaMKII (1:1000, #3362), phospho-CaMKII (Thr-286) (1:1000, #3361), Bcl-2 (1:1000, #3498), and Bax (1:1000, #2772) were purchased from Cell Signaling Technology (Boston, MA, USA). The primary antibodies against p-nuclear factor of activated T-cells c3 (NFATc3) (Ser-265) (1:250, sc-32982), NFATc3 (1:250, sc-8321), p91phox (1:1000, sc-130543), 3- nitrotyrosine

(3-NT) (1:250, sc-32757) and 3-glyceraldehyde-3-phosphate dehydrogenase (GAPDH) (1:200, sc-25778) were purchased from Santa Cruz Biotechnology (Santa Cruz, CA, USA) and the primary antibodies against phospho-ryanodine receptor 2 (RyR2) (1:5000) (Ser-2814, #A010-31) and phospho-RyR2 (1:5000) (Ser-2808, A010-30) were purchased from Badrilla (Leeds, UK). The primary antibody against p22phox (1:1000, ab75941) was purchased from Abcam (Cambridge, UK) and RyR2 (1:1000, #MA3–916) was purchased from Thermo Fisher (Rockland, IL, USA). The primary and secondary antibodies were diluted in Tris-buffered saline (pH 7.6) with 0.1% Tween 20 and 5% bovine serum albumin. Blots were visualized with enhanced chemiluminescence solution (ECL; Prime Western Blotting Detection Reagent, GE, Healthcare, Piscataway, NJ, USA) and scanned with a densitometer (LAS-1000, Fuji Photo Film, Tokyo, Japan). The amount of expression in the control was taken as 100% in each determination, in accordance with previous studies [25,30]. The reason why there are different numbers of samples in different western blotting figures (Fig 2C–2D, Fig 3C, Fig 4C, Fig 5 and Fig 6) is that we excluded outliers (extremely low or high values, compared to others in the same groups) using the Smirnov-Grubbs test [31].

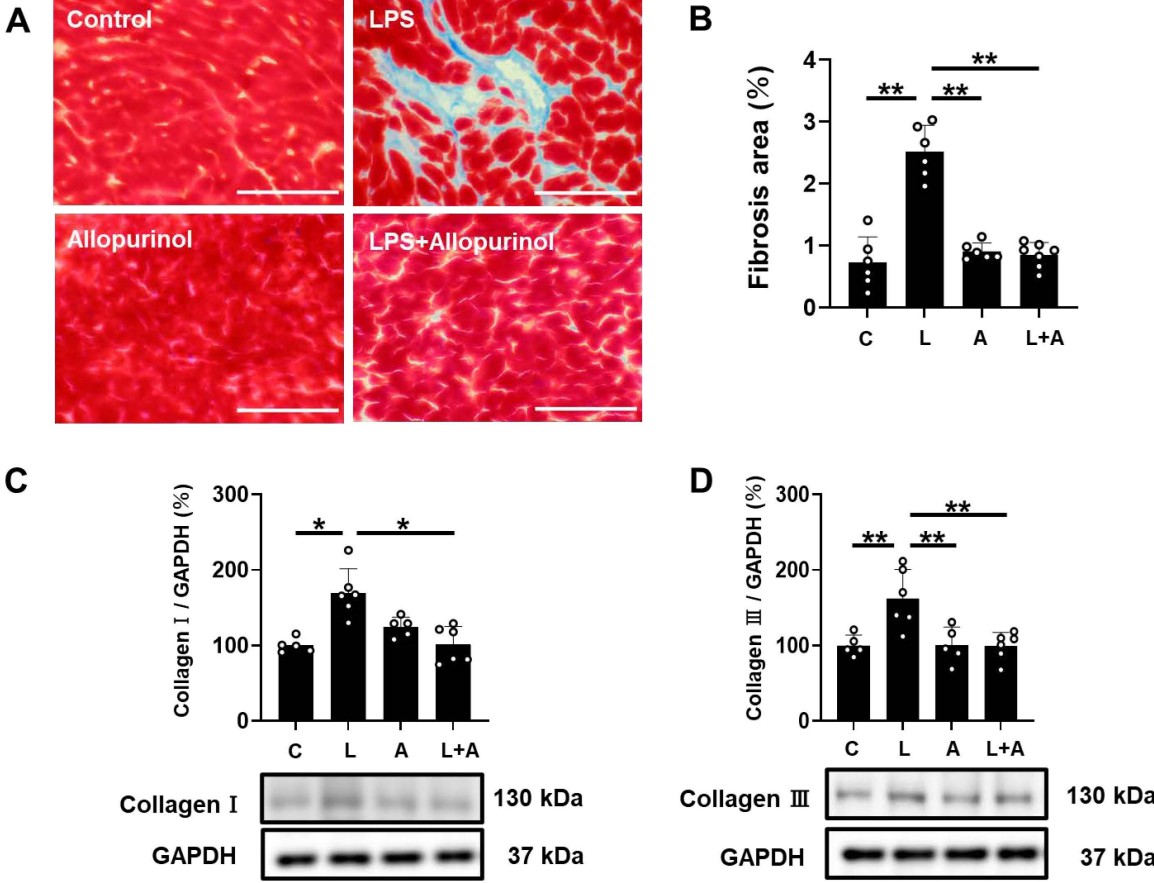

**Fig 2. Effects of allopurinol on PG-LPS-induced fibrosis in cardiac muscle. (A)** Representative images of Masson-trichrome-stained sections of cardiac muscle in the Control (*upper left*), PG-LPS (*upper right*), allopurinol (*lower left*), and PG-LPS + allopurinol (*lower right*) groups. Scale bar: 100 μm **(B)** The area of fibrosis was significantly increased in the PG-LPS group ($n = 6$, **$P < 0.01$), but this increase was blocked in the PG-LPS + allopurinol group ($n = 7$, **$P < 0.01$) by ANOVA/Tukey-Kramer. **(C-D)** Expression of collagen I **(C)** and collagen III **(D)** was significantly increased in the PG-LPS group, but these increases were blocked in the PG-LPS + allopurinol group ($n = 6$ each). *$P < 0.05$, **$P < 0.01$ by non-parametric ANOVA/Steel-Dwass **(C)** or ANOVA/Tukey-Kramer **(D)**. Data are presented as mean ± SD and dots show individual data. Images of full-size immunoblots are presented in S1 and S2 Fig **of** S1 Data.

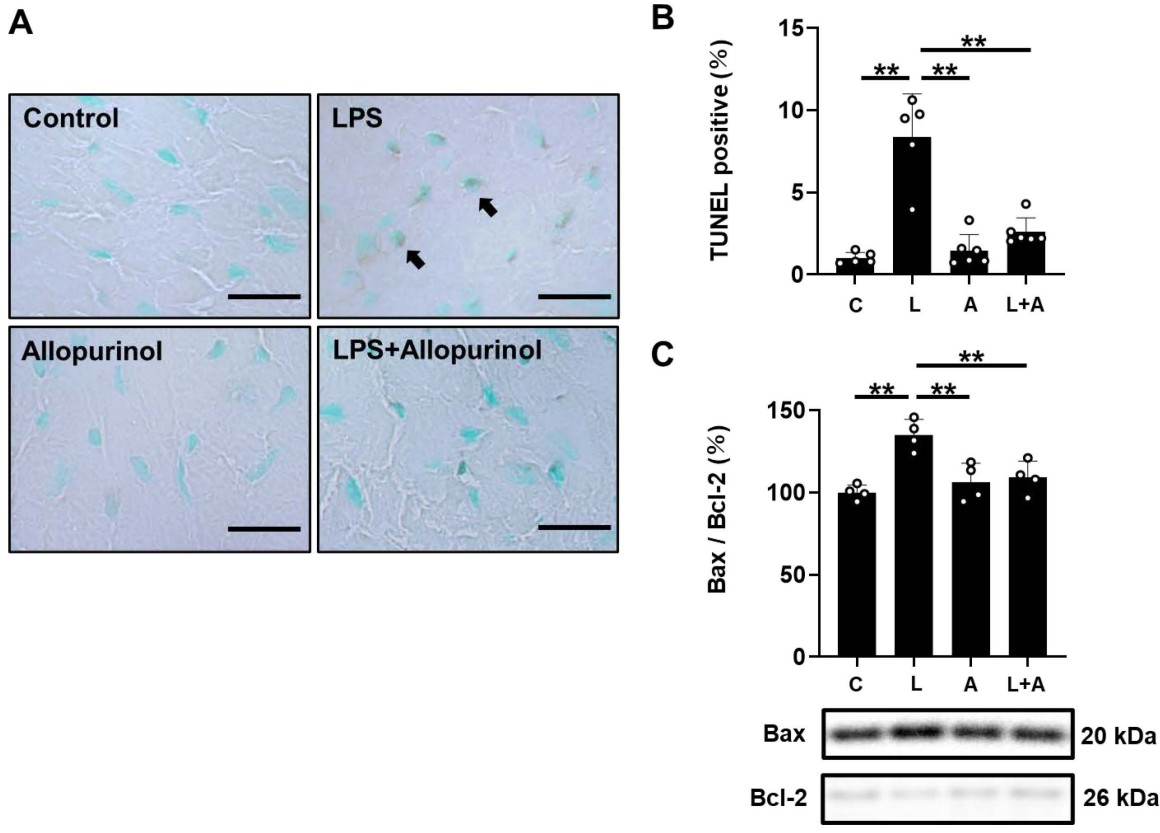

**Fig 3. Effects of allopurinol on cardiac myocyte apoptosis induced by chronic PG-LPS infusion.** (A) Representative images of TUNEL-stained sections of cardiac muscle from the Control (*upper left*), PG-LPS (LPS) (*upper right*), allopurinol (*lower left*) and PG-LPS + allopurinol (LPS + Cap) (*lower right*) groups. Scale bars: 2 μm. (**B**) The number of TUNEL-positive cardiac myocytes ~~area of fibrosis~~ was significantly increased in the PG-LPS group (L) (*P* < 0.01 vs. Control), and this increase was significantly attenuated by allopurinol (L + A). **P* < 0.01 vs. Control (C) or **P* < 0.01 vs. PG-LPS group (L) by ANOVA/Tukey-Kramer. (**C**) The Bax/Bcl-2 ratio was significantly increased in the PG-LPS group (*n* = 4), but this increase was blocked in the PG-LPS + allopurinol group (*n* = 4). **P* < 0.01 by ANOVA/Tukey-Kramer. Data are presented as mean ± SD and dots show individual data. Images of full-size immunoblots are presented in S3 Fig of S1 Data.

## Statistical analysis

Data are presented as means ± standard deviation (SD). The Shapiro-Wilk test was performed to evaluate if the sample showed a normal distribution (S3 Data) [32]. When the distribution was not normal, we used a non-parametric test for analysis (Steel-Dwass test) [Fig 1C, Fig 2C and Table 1 (**LVPWTd**)]. Comparisons were performed using one-way ANOVA followed by the Tukey-Kramer *post hoc* test (hereafter abbreviated as ANOVA/Tukey-Kramer) (Fig 1B, 1D, 1E, Fig 2B, 2D, Fig 3B, 3C, Fig 4B, 4C, Fig 5, Table 1 and S1 Fig in S1 Data of S2 Data) or non-parametric one-way ANOVA followed by the Steel-Dwass *post hoc* test (hereafter abbreviated as non-parametric ANOVA/Steel-Dwass) [Fig 1C, Fig 2C and Table 1 (**LVPWTd**)] [33]. Differences were considered significant when *P* < 0.05.

## Results

### Effects of PG-LPS on body weight and size of heart, lung and liver with/without allopurinol

PG-LPS treatment for a week did not alter the body weight of the control, PG-LPS, allopurinol and PG-LPS + allopurinol groups at 1 week after the PG-LPS treatment (Fig 1B), and the

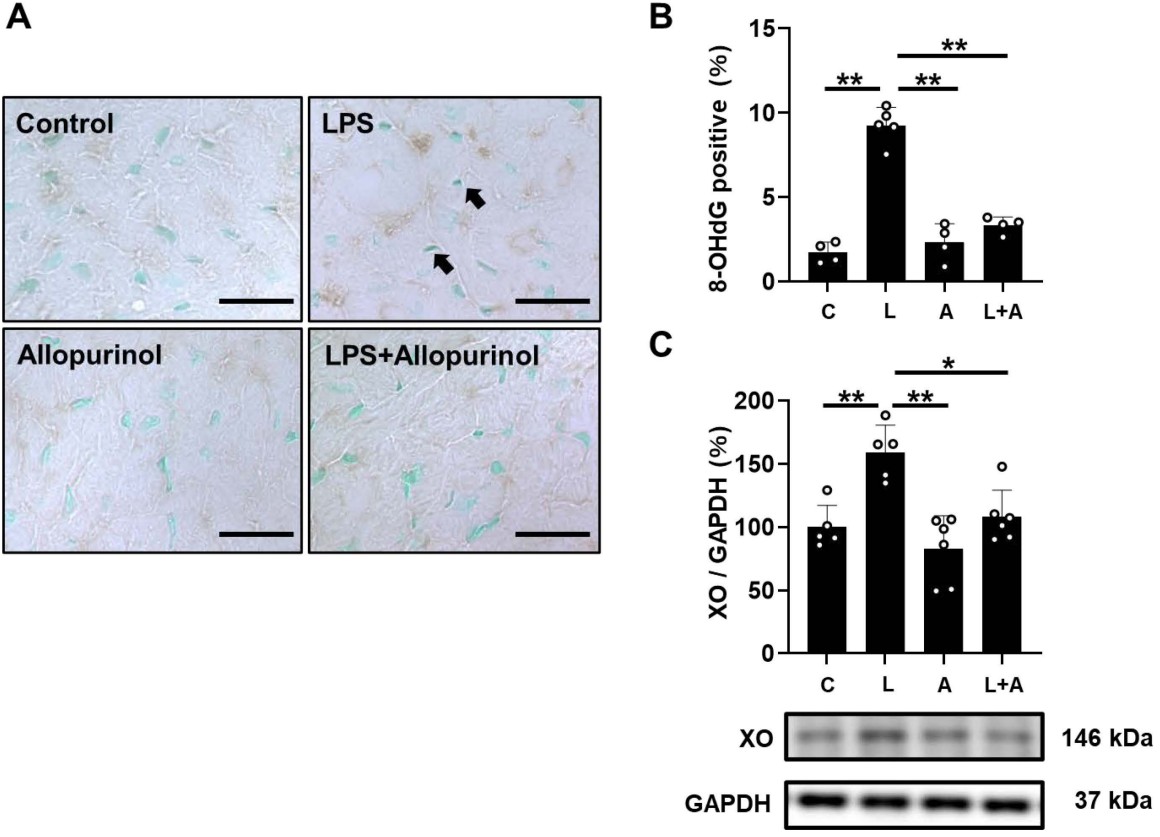

**Fig 4. Effects of allopurinol on chronic PG-LPS-induced oxidative stress in cardiac muscle.** (A) Representative immunohisto-chemical images of oxidative DNA damage (8-OHdG) in cardiac muscle from the Control (*upper left*), PG-LPS (LPS) (*upper right*), allopurinol (*lower left*) and PG-LPS + allopurinol (L + A) (*lower right*) groups. Scale bars: 2 μm (B) 8-OHdG-positive nuclei were significantly increased in the PG-LPS group ($n = 5$), but this increase was blocked in the PG-LPS + allopurinol group (L + A) ($n = 4$). $^{**}P$ < 0.01 by ANOVA/Tukey-Kramer. **T** (C) Expression of XO was significantly increased in the PG-LPS group ($n = 5$), and this increase was significantly blocked in the PG-LPS + allopurinol group (L + A) ($n = 6$). $^{*}P$ < 0.05 by ANOVA/Tukey-Kramer. Data are presented as mean ± SD and dots show individual data. Images of full-size immunoblots are presented in S4 Fig of S1 Data.

consumed amounts of both food and water were also similar among the four groups (S1 Fig in **S1 Data** of S2 Data).

We also examined the effects of PG-LPS with/without allopurinol on heart size in terms of cardiac muscle mass per tibial length ratio (mg/mm) (Fig 1C), as well as the effects on wet lung and liver mass per tibial length ratio (Fig 1D and 1E). Similar results were obtained among the four groups.

Thus, neither PG-LPS nor allopurinol at the dose used in this experiment appeared to influence growth, food/water consumption, cardiac hypertrophy, lung edema or liver congestion during the 1-week experimental period.

### Effects of PG-LPS on cardiac function with/without allopurinol treatment

We also conducted echocardiography (Table 1) to evaluate cardiac function in terms of left ventricular ejection fraction (EF) and fractional shortening (%FS). Both parameters were significantly decreased in the PG-LPS-treated group [EF: Control ($n = 6$) vs. PG-LPS ($n = 7$): 68 ± 1.3 vs. 60 ±2.7%, $P$ < 0.01; %FS: 33 ± 0.9 vs. 28 ± 1.6%, $P$ < 0.01 by ANOVA/Tukey-Kramer]. Allopurinol alone ($n = 6$) had no effect on EF or %FS, but blocked the PG-LPS-induced decrease of EF and %FS at 1 week [EF: PG-LPS ($n = 7$) vs. PG-LPS + allopurinol ($n =$

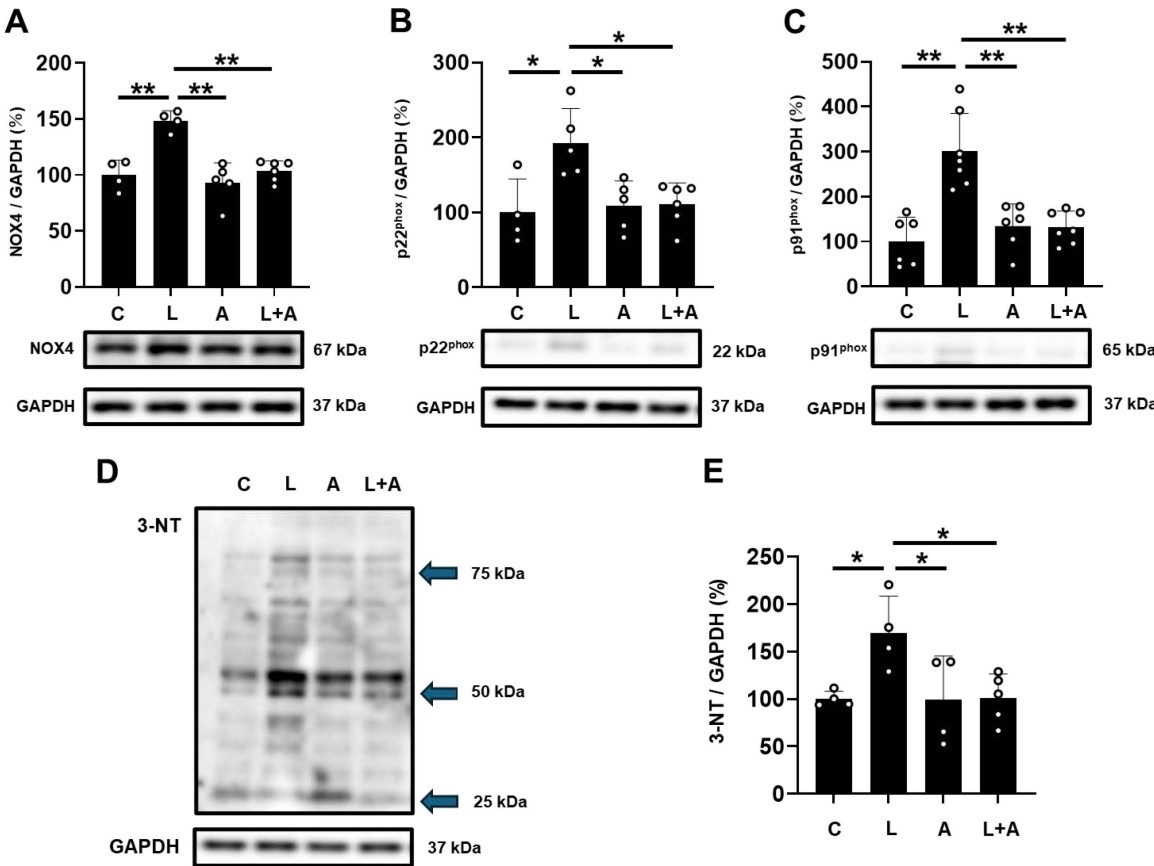

**Fig 5. Effects of allopurinol on PG-LPS-induced increases in NOX4 , p22$^{phox}$, p91$^{phox}$ and 3-NT in cardiac muscle.** (A) NOX4 expression was significantly increased in the PG-LPS group (L), and this increase was significantly attenuated in the PG-LPS + allopurinol group (L + A). $^{**}P < 0.01$ vs. Control (C) or $^{**}P < 0.01$ vs. L + A by ANOVA/Tukey-Kramer. Images of full-size immunoblots are shown in S5 Fig of S1 Data. **(B)** p22$^{phox}$ expression was significantly increased in the PG-LPS group (L), and this increase was significantly attenuated in the PG-LPS + allopurinol group (L + A). $^{*}P < 0.05$ vs. Control (C) or $^{*}P < 0.05$ vs. L + A by ANOVA/Tukey-Kramer. Images of full-size immunoblots are shown in S6 Fig of S1 Data. **(C)** p91$^{phox}$ expression was significantly increased in the PG-LPS group (L), and this increase was significantly attenuated in the PG-LPS + allopurinol group (L + A). $^{**}P < 0.01$ vs. Control (C) or $^{**}P < 0.01$ vs. L + A by ANOVA/Tukey-Kramer. Images of full-size immunoblots are shown in S7 Fig of S1 Data. (D) Representative immunoblot showing expression levels of 3-NT in cardiac muscle from the Control (C), PG-LPS (L), allopurinol (A) and PG-LPS + allopurinol (L + A) groups. Images of full-size immunoblots are shown in S8 Fig of S1 Data. (E) 3-NT expression was significantly increased in the PG-LPS group (L), and this increase was significantly attenuated in the PG-LPS + allopurinol group (L + A). $^{*}P < 0.05$ vs. Control (C) or $^{*}P < 0.05$ vs. L + A by ANOVA/Tukey-Kramer.

7): 60 ± 2.7 vs. 67 ± 1.1%, $P < 0.01$; %FS: PG-LPS ($n = 7$) vs. PG-LPS + allopurinol ($n = 7$): 28 ± 1.6 vs. 33 ± 0.8%, $P < 0.01$; ANOVA/Tukey-Kramer].

These data suggest that PG-LPS treatment decreases cardiac dysfunction at least in part via the increase of ROS production.

## Effects of PG-LPS on cardiac fibrosis with/without allopurinol treatment

We examined the effects of PG-LPS with/without allopurinol on fibrosis in cardiac muscle by means of Masson-trichrome staining (Fig 2A). PG-LPS treatment significantly increased the area of fibrosis in cardiac muscle [Control ($n = 6$) vs. PG-LPS ($n = 6$): 0.7 ± 0.4 vs. 2.5 ± 0.4%, $P < 0.01$; ANOVA/Tukey-Kramer] (Fig 2B), in accordance with our previous findings [16, 17]. Allopurinol alone did not alter the area of fibrosis, but blocked the PG-LPS-induced increase

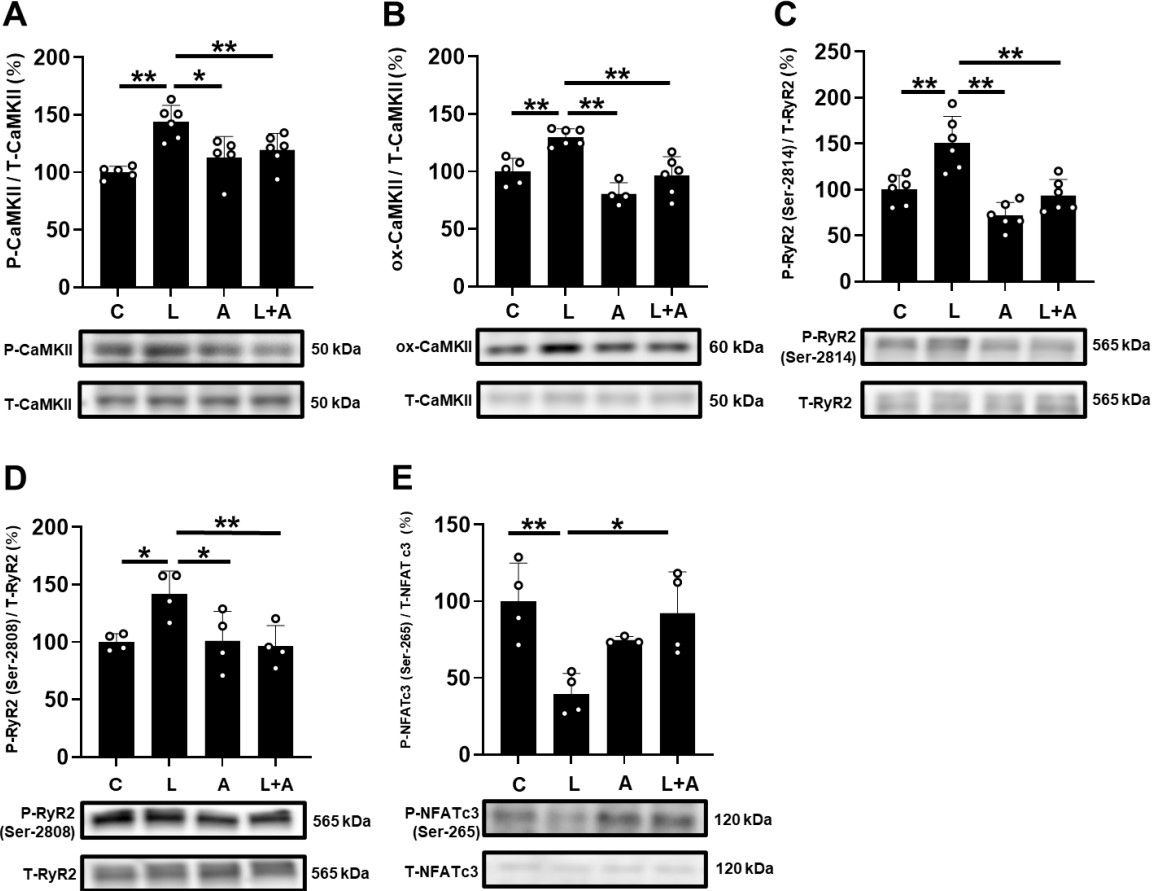

**Fig 6. Effects of allopurinol on PG-LPS-induced phospho-CaMKII , ox-CaMKII, phospho-RyR2 (Ser-2814), phospho-RyR2 (Ser-2808) and phospho-NFATc3 in cardiac muscle.** (A) CaMKII phosphorylation (Thr-286) was significantly increased in the PG-LPS group (L), and this increase was significantly attenuated in the PG-LPS + allopurinol group (L + A). $^{**}P < 0.01$ vs. Control (C) or $^{*}P < 0.05$ vs. **L** + A by ANOVA/Tukey-Kramer. Images of full-size immunoblots are shown in S9 Fig of <u>S1 Data</u>. (B) CaMKII oxidization (methionine-281/282) was significantly increased in the PG-LPS group (L), and this increase was significantly attenuated in the PG-LPS + allopurinol group (L + A). $^{**}P < 0.01$ vs. Control (C) or $^{**}P < 0.01$ vs. **L** + A by ANOVA/Tukey-Kramer. Images of full-size immunoblots are shown in S10 Fig of <u>S1 Data</u>. (C) RyR2 phosphorylation (Ser-2814) was significantly increased in the PG-LPS group (L), and this increase was significantly attenuated in the PG-LPS + allopurinol group (L + A). $^{**}P < 0.01$ vs. Control (C) or $^{**}P < 0.01$ vs. **L** + A by ANOVA/Tukey-Kramer. Images of full-size immunoblots are shown in S11 Fig of <u>S1 Data</u>. (D) RyR2 phosphorylation (Ser-2808) was significantly increased in the PG-LPS group (L), and this increase was significantly attenuated in the PG-LPS + allopurinol group (L + A). $^{*}P < 0.05$ vs. Control (C) or $^{**}P < 0.01$ vs. **L** + A by ANOVA/Tukey-Kramer. Images of full-size immunoblots are shown in S12 Fig of <u>S1 Data</u>. (E) NFATc3 phosphorylation (Ser-265) was significantly increased in the PG-LPS group (L), and this increase was significantly attenuated in the PG-LPS + allopurinol group (L + A). $^{**}P < 0.01$ vs. Control (C) or $^{*}P < 0.05$ vs. **L** + A by ANOVA/Tukey-Kramer. Images of full-size immunoblots are shown in S13 Fig of <u>S1 Data</u>.

of fibrosis [PG-LPS ($n = 6$) vs. PG-LPS + allopurinol ($n = 7$): 2.5 ± 0.4 vs. 0.9 ± 0.2%, $P < 0.01$; ANOVA/Tukey-Kramer].

These data suggest that cardiac fibrosis induced by PG-LPS might be mediated, at least in part through the increase of ROS production.

## Effects of PG-LPS on collagen 1 and 3 protein expression with/without allopurinol treatment

We examined the protein expression of collagen I (<u>Fig 2C</u>) and collagen III (<u>Fig 2D</u>) in the heart in the four groups. The expression levels were significantly increased in the heart of

**Table 1. Cardiac function assessed by echocardiography with/without allopurinol.**

|  | C | L | A | L + A |
|---|---|---|---|---|
| n | 6 | 7 | 6 | 7 |
| EF | 68 ± 1.3 | 60 ± 2.7 ** | 67 ± 2.2 | 67 ± 1.1 ## |
| EDV | 0.2 ± 0.03 | 0.2 ± 0.02 | 0.20 ± 0.01 | 0.20 ± 0.01 |
| ESV | 0.07 ± 0.006 | 0.08 ± 0.011 | 0.06 ± 0.007 ## | 0.06 ± 0.004 ## |
| %FS | 33 ± 0.9 | 28 ± 1.6 ** | 33 ± 1.6 | 33 ± 0.8 ## |
| LVIDd | 4.5 ± 0.1 | 4.4 ± 0.1 | 4.3 ± 0.08 | 4.3 ± 0.07 |
| LVIDs | 3.0 ± 0.08 | 3.2 ± 0.2 | 2.9 ± 0.1 ## | 2.9 ± 0.06 ## |
| HR | 420 ± 35 | 414 ± 45 | 402 ± 27 | 416 ± 37 |
| SV | 0.15 ± 0.011 | 0.12 ± 0.006 ** | 0.13 ± 0.004 * | 0.13 ± 0.006 * |
| CO | 63 ± 2.6 | 52 ± 6.5 * | 54 ± 4.1 * | 56 ± 6.4 |
| IVSTd | 0.5 ± 0.07 | 0.4 ± 0.05 | 0.4 ± 0.03 | 0.5 ± 0.03 |
| IVSTs | 0.9 ± 0.07 | 0.8 ± 0.03 ** | 0.9 ± 0.04 | 0.9 ± 0.04 * |
| LVPWTd | 0.53 ± 0.05 | 0.52 ± 0.06 | 0.47 ±.05 | 0.48 ± 0.04 |
| LVPWTs | 0.96 ± 0.06 | 0.86 ± 0.04 * | 0.89 ± 0.08 | 0.92 ± 0.05 |

EF (%): left ventricular ejection fraction.

EDV (mL): left ventricular end-diastolic volume.

ESV (mL): left ventricular end-systolic volume.

%FS: % fractional shortening.

LVIDd (mm): left ventricular internal dimension at end-diastole.

LVIDs (mm): left ventricular internal dimension at end-systole.

HR (bpm): heart rate.

SV (mL): stroke volume.

CO (mL/min): cardiac output.

IVSTd (mm): interventricular septum thickness at end-diastole.

LVSTs (mm): interventricular septum thickness at end-systole

LVPWTd (mm): left ventricular posterior wall thickness at end-diastole. LVPWTs (mm).

LVPWTs (mm): left ventricular posterior wall thickness at end-systole.

**$P < 0.01$ vs. Control by Tukey/Kramer.

*$P < 0.05$ vs. Control by Tukey/Kramer.

##$P < 0.01$ vs. LPS by Tukey/Kramer.

#$P < 0.05$ vs. LPS by Tukey/Kramer.

C: control, L: PG-LPS, A: allopurinol, L + A: PG-LPS + allopurinol.

PG-LPS-treated mice [collagen I: Control ($n = 5$) vs. PG-LPS ($n = 6$): 100 ± 9.4 vs. 170 ± 32%, $P < 0.05$; non-parametric ANOVA/Steel-Dwass; collagen III: Control ($n = 5$) vs. PG-LPS ($n = 6$): 100 ± 14 vs. 162 ± 39%, $P < 0.01$; ANOVA/Tukey-Kramer] (Fig 2C and 2D). Allopurinol alone did not alter the expression of collagen I or collagen III, but blocked the PG-LPS-induced increases [collagen I: PG-LPS ($n = 6$) vs. PG-LPS + allopurinol ($n = 6$): 170 ± 32 vs. 101 ± 24%, $P < 0.01$; ANOVA/Tukey-Kramer; collagen III: PG-LPS ($n = 6$) vs. PG-LPS + allopurinol ($n = 6$): 162 ± 39 vs. 99 ± 18%, $P < 0.01$; ANOVA/Tukey-Kramer].

These data suggest that increased protein expression of collagen I and collagen III induced by PG-LPS might be mediated at least in part through the increase of ROS production.

## Effects of PG-LPS on cardiac apoptosis with/without allopurinol treatment

We next evaluated cardiac apoptosis in PG-LPS-treated mice with/without allopurinol treatment by means of terminal deoxyribonucleotidyl transferase (TdT)-mediated biotin-16-deoxyuridine triphosphate (dUTP) nick-end labeling (TUNEL) (Fig 3A).

We first prepared positive and negative control sections by incubating cells with (positive control) or without (negative control) DNaseI for 15 min at 37°C and confirmed that the TUNEL staining procedure could clearly discriminate TUNEL-positive and non-positive nuclei (S2 Fig in **S1 Data** of S2 Data).

PG-LPS treatment significantly increased cardiac myocyte apoptosis [Control ($n = 5$) vs. PG-LPS ($n = 5$): 1.0 ± 0.3 vs. 8.3 ± 2.6%, $P < 0.01$ vs. Control; ANOVA/Tukey-Kramer]. Allopurinol alone had no effect on the number of TUNEL-positive cardiac myocytes, but blocked the PG-LPS-induced increase of TUNEL-positive cardiac myocytes [PG-LPS ($n = 5$) vs. PG-LPS + allopurinol ($n = 6$): 8.3 ± 2.6 vs. 2.6 ± 0.8%, $P < 0.01$ vs. PG-LPS; ANOVA/Tukey-Kramer] (Fig 3B).

We also examined the effects of PG-LPS on the ratio of Bcl-2 associated X protein (Bax), an accelerator of apoptosis, to B cell lymphoma 2 (Bcl-2), a regulator of apoptosis, in the heart (Fig 3C) and found that it was significantly increased in the heart of PG-LPS-treated mice [Control ($n = 4$) vs. PG-LPS ($n = 4$): 100 ± 4.6 vs. 135 ± 9.4%, $P < 0.01$ vs. Control; ANOVA/Tukey-Kramer]. However, the increase was blocked by allopurinol [PG-LPS ($n = 4$) vs. PG-LPS + allopurinol ($n = 4$): 135 ± 9.4 vs. 109 ± 10%, $P < 0.01$ vs. PG-LPS; ANOVA/Tukey-Kramer] (Fig 3C).

## Effects of PG-LPS on oxidative stress with/without allopurinol treatment

We evaluated oxidative stress in the myocardium by means of 8-hydroxy-2'-deoxyguanosine (8-OHdG) immunostaining (Fig 4A).

We first prepared positive and negative control sections by incubating cells with (positive control) or without (negative control) 0.3% $H_2O_2$ in TBS-T for 1 h at room temperature before the anti-OHdG antibody treatment and confirmed that the 8-OHdG staining procedure could clearly discriminate 8-OHdG-positive and non-positive nuclei (S3 Fig in **S1 Data** of S2 Data).

The ratio of 8-OHdG-positive/total cardiac myocytes was significantly increased in the PG-LPS-treated mice [Control ($n = 4$) vs. PG-LPS ($n = 5$): 1.7 ± 0.6 vs. 9.2 ± 1.1%, $P < 0.01$ vs. Control; ANOVA/Tukey-Kramer], and the increase was suppressed by allopurinol [PG-LPS ($n = 5$) vs. PG-LPS + allopurinol ($n = 4$): 9.2 ± 1.1 vs. 3.3 ± 0.5%, $P < 0.01$ vs. PG-LPS; ANOVA/Tukey-Kramer] (Fig 4B).

We also examined XO expression in the heart in the four groups. XO expression was significantly increased in the PG-LPS-treated group [Control ($n = 5$) vs. PG-LPS ($n = 5$): 100 ± 17 vs. 159 ± 21%, $P < 0.01$ vs. Control; ANOVA/Tukey-Kramer]. However, this increase was significantly inhibited by allopurinol [PG-LPS ($n = 5$) vs. PG-LPS + allopurinol ($n = 6$): 159 ± 21 vs. 108 ± 21%, $P < 0.05$ vs. PG-LPS; ANOVA/Tukey-Kramer] in accordance with the previous study in animals with heart failure [34] or atrial remodeling [35] treated with allopurinol (Fig 4C).

## Effects of PG-LPS on NOX4 expression with/without allopurinol

TLR4 stimulation causes cardiac ROS generation through a number of pathways, including nicotinamide adenine dinucleotide phosphate oxidase 4 (NOX4), as established by us [16] and another group [36]. Two NOX isoforms, NOX2 and NOX4, are expressed in the heart, and their activity is regulated by their expression levels [37, 38]. It has been shown that allopurinol treatment reduces oxidative stress by decreasing the expression level of Nox4, in addition to XO, and might block aortic aneurysm in a mouse model of Marfan syndrome [39].

Importantly, we previously demonstrated that PG-LPS-induced cardiac fibrosis and remodeling might be caused by ROS production via TLR4/NOX 4 interaction [16]. Therefore, we compared NOX4 protein expression in the heart among the four groups (Fig 5A). NOX4 expression was significantly increased in the PG-LPS-treated group [Control ($n = 4$) vs.

PG-LPS ($n = 4$): 100 ± 13 vs. 148 ± 8.9%, $P < 0.01$ vs. Control; ANOVA/Tukey-Kramer], and the increase was suppressed by allopurinol [PG-LPS ($n = 4$) vs. PG-LPS + allopurinol ($n = 6$): 148 ± 8.9 vs. 104 ± 9.0%, $P < 0.01$ vs. PG-LPS; ANOVA/Tukey-Kramer].

These data, together with the results in Fig 4C, suggest that PG-LPS-induced ROS production might contribute at least in part to the upregulation of NOX4 and XO.

### Effects of PG-LPS on p22$^{phox}$ expression with/without allopurinol

ROS production by Nox4 is dependent on the p22$^{phox}$ protein expression level [40, 41]. We thus examined the expression of p22$^{phox}$ protein in the heart among the four groups and found that p22$^{phox}$ expression was significantly increased in the PG-LPS-treated group [Control ($n = 4$) vs. PG-LPS ($n = 5$): 100 ± 45 vs. 183 ± 47%, $P < 0.05$ vs. Control; ANOVA/Tukey-Kramer]. This increase was suppressed by allopurinol [PG-LPS ($n = 5$) vs. PG-LPS + allopurinol ($n = 6$): 183 ± 47 vs. 111 ± 28%, $P < 0.05$ vs. PG-LPS; ANOVA/Tukey-Kramer] (Fig 5B).

These data suggest that PG-LPS-induced ROS production might enhance the activity of Nox4 by upregulating the expression of p22$^{phox}$, leading to increased ROS production via Nox4, while allopurinol treatment might decrease Nox4-dependent ROS production through a decrease of p22$^{phox}$ expression.

### Effects of PG-LPS on the expression of p91$^{phox}$ and 3-NT with/without allopurinol

p91$^{phox}$ and 3-NT are markers of oxidative stress and protein oxidation damage [42, 43]. We thus examined the expression levels of p91$^{phox}$ and 3-NT in the heart among the four groups and found that both were significantly increased in the PG-LPS-treated group [p91$^{phox}$: Control ($n = 6$) vs. PG-LPS ($n = 7$): 100 ± 54 vs. 301 ± 82%, $P < 0.01$ vs. Control; 3-NT: Control ($n = 4$) vs. PG-LPS ($n = 4$): 100 ± 8 vs. 170 ± 39%, $P < 0.05$ vs. Control; ANOVA/Tukey-Kramer each]. Further, these increases were suppressed by allopurinol [p91$^{phox}$: PG-LPS ($n = 7$) vs. PG-LPS + allopurinol ($n = 7$): 301 ± 82 vs. 132 ± 36%, $P < 0.01$ vs. PG-LPS; 3-NT: PG-LPS ($n = 4$) vs. PG-LPS + allopurinol ($n = 5$): 170 ± 39 vs. 101 ± 26%, $P < 0.05$ vs. PG-LPS; ANOVA/Tukey-Kramer] (Fig 5C-5E).

These data also support the view that allopurinol has a protective action against oxidative stress induced by PG-LPS.

### Effects of PG-LPS on CaMKII phosphorylation and oxidation with/without allopurinol

CaMKII is activated via phosphorylation and oxidation in the presence of ROS and contributes to the development of cardiac remodeling and dysfunction induced by PG-LPS [16]. We thus examined the amounts of phospho-CaMKII (Thr-286) (Fig 6A) and oxidized methionine-281/282 CaMKII (ox-CaMKII) (Fig 6B) in the heart in the four groups and found that they were significantly increased in the PG-LPS-treated group [p-CaMKII: Control ($n = 5$) vs. PG-LPS ($n = 6$): 100 ± 5.2 vs. 144 ± 14%; ox-CaMKII: Control ($n = 5$) vs. PG-LPS ($n = 6$): 100 ± 11 vs. 130 ± 7.4%, $P < 0.01$ vs. Control; ANOVA/Tukey-Kramer]. The increase was suppressed by allopurinol [p-CaMKII: PG-LPS ($n = 6$) vs. PG-LPS + allopurinol ($n = 6$): 144 ± 14 vs. 119 ± 14%, $P < 0.01$ vs. PG-LPS; ox-CaMKII: PG-LPS ($n = 6$) vs. PG-LPS + allopurinol ($n = 6$): 130 ± 4.4 vs. 96 ± 17%, $P < 0.01$ vs. PG-LPS; ANOVA/Tukey-Kramer].

### Effects of PG-LPS on RyR2 phosphorylation with/without allopurinol

Since phosphorylation of most Ca$^{2+}$-handling proteins is altered in many models of experimental heart failure [25,36], which might lead to increased Ca$^{2+}$ leakage, we compared the

effects of PG-LPS on RyR2 phosphorylation at Ser-2814 (Fig 6C) and Ser-2808 (Fig 6D). These phosphorylations are mediated by CaMKII and protein kinase A, respectively [44].

Phospho-RyR2 (Ser-2814) was significantly increased in the heart of PG-LPS-treated mice [Control ($n$ = 6) vs. PG-LPS ($n$ = 6): 100 ± 16 vs. 151 ± 29%, $P$ < 0.01 vs. Control; ANOVA/ Tukey-Kramer]. Again, this increase was significantly attenuated by allopurinol [PG-LPS ($n$ = 6) vs. PG-LPS + allopurinol ($n$ = 6): 151 ± 29 vs. 93 ± 17%, $P$ < 0.01 vs. PG-LPS; ANOVA/ Tukey-Kramer] (Fig 6C).

Phospho-RyR2 (Ser-2808) was also significantly increased in the heart of PG-LPS-treated mice [Control ($n$ = 4) vs. PG-LPS ($n$ = 4): 100 ± 7.2 vs. 142 ± 20%, $P$ < 0.05 vs. Control; ANOVA/Tukey-Kramer], and this increases was significantly attenuated by allopurinol [PG-LPS ($n$ = 4) vs. PG-LPS + allopurinol ($n$ = 4): 142 ± 20 vs. 96 ± 18%, $P$ < 0.01 vs. PG-LPS; ANOVA/Tukey-Kramer] (Fig 6D).

These data suggest that PG-LPS might increase RyR2 phosphorylation on Ser-2808 and Ser-2814, at least in part through the activation of the TLR4-Nox4 signaling pathway.

### Effects of NFATc3 phosphorylation with/without allopurinol

The maintenance of calcium ($Ca^{2+}$) homeostasis during muscle contraction is requisite for optimal contractile function, and altered $Ca^{2+}$ homeostasis might induce hyperactivation of calcineurin-NFAT signaling [45]. We thus examined the activation level of calcineurin-NFAT signaling in terms of the phosphorylation level on serine 265 of NFATc3, which is a $Ca^{2+}$-handling protein involved in $Ca^{2+}$ homeostasis in cardiac muscle (Fig 6E). We found that the phosphorylation level was significantly decreased in the PG-LPS-treated group (Control [$n$ = 4] vs. PG-LPS [$n$ = 4]: 100 ± 26 vs. 40 ± 14%, $P$ < 0.01 vs. Control; ANOVA/Tukey-Kramer). Further, this decrease was suppressed by allopurinol (PG-LPS [$n$ = 4] vs. PG-LPS + allopurinol [$n$ = 4]: 40 ± 14 vs. 94 ± 28%, $P$ < 0.05 vs. PG-LPS; ANOVA/Tukey-Kramer) (Fig 6E).

These data indicated that allopurinol might protect the heart from $Ca^{2+}$ handling impairment induced by PG-LPS.

## Discussion

Numerous cross-sectional, case-control and cohort epidemiological studies have demonstrated that periodontitis is associated with CVD, independently of confounding factors such as smoking and obesity [46–48]. Moreover, clinical interventional studies indicate that treatment of periodontitis reduces systemic inflammation and has favorable effects on subclinical markers of CVD, although the molecular mechanisms involved remain elusive [48–52].

Our findings here indicate that cardiac function was significantly impaired in mice treated with PG-LPS at a dose consistent with circulating levels in periodontitis patients, and myocyte apoptosis, fibrosis and oxidative stress were significantly increased. Importantly, these changes were blunted by the XO inhibitor allopurinol. We then examined the mechanism of these changes.

Patients with periodontitis exhibit accelerated purine degradation and enhanced XO expression in the periodontium [53, 54]. Enhanced secretion of uric acid has been observed in immune cells stimulated by periodontal pathogens [54, 55], and in the gingiva of mice with periodontitis [56]. Given that XO is ubiquitous, and is sensitive to inflammation and oxidative stress [57], periodontitis-induced low-grade systemic inflammation may accelerate purine metabolism in distant organs. We hypothesized that the purine degradation pathway might be increased not only in the periodontium, but also in the heart. This hypothesis was confirmed by the observation of up-regulated expression of XO in the heart of PG-LPS-treated mice. We also showed that this increase was attenuated by allopurinol.

The effect of allopurinol on XO expression level is controversial: some studies have found that allopurinol inhibits XO activity without reducing its expression level [58, 59], while others have found a reduction of its expression level [34, 35].

Patients with periodontitis are at high risk for CVD, which might be due to increased ROS production, as shown by us [16, 17] and others [60]. We have recently demonstrated that expression of XO and NOX4 is significantly increased in PG-LPS-treated mice, as used in this study, and these increases were suppressed by the angiotensin converting enzyme inhibitor captopril [17]. Thus, the present and previous findings suggest that another important source of ROS in the cardiovascular system in patients with periodontitis might be NOX4, besides XO [17]. In addition, the renin-angiotensin system (RAS) might contribute at least in part to the increase of ROS via XO and NOX4 in the heart, leading to the development of cardiac remodeling and dysfunction in PG-LPS-treated mice (Fig 7).

We recently demonstrated that persistent subclinical exposure to PG-LPS in mice induces cardiac dysfunction, myocyte apoptosis and fibrosis [16]. In addition, co-treatment with the TLR4 inhibitor TAK-242 alleviates the cardiac dysfunction and remodeling induced by PG-LPS. [16]. TLR4 activation was recently demonstrated to be involved in activation of the RAS induced by uric acid in adipose tissue, causing hypertension and increased expression of inflammatory cytokines [61]. In addition, the interplay of TLR4 signaling and the RAS might contribute to the pathogenesis of cardiovascular changes in Fabry disease [62]. Our findings in this and previous studies support the hypothesis that cardiac dysfunction induced

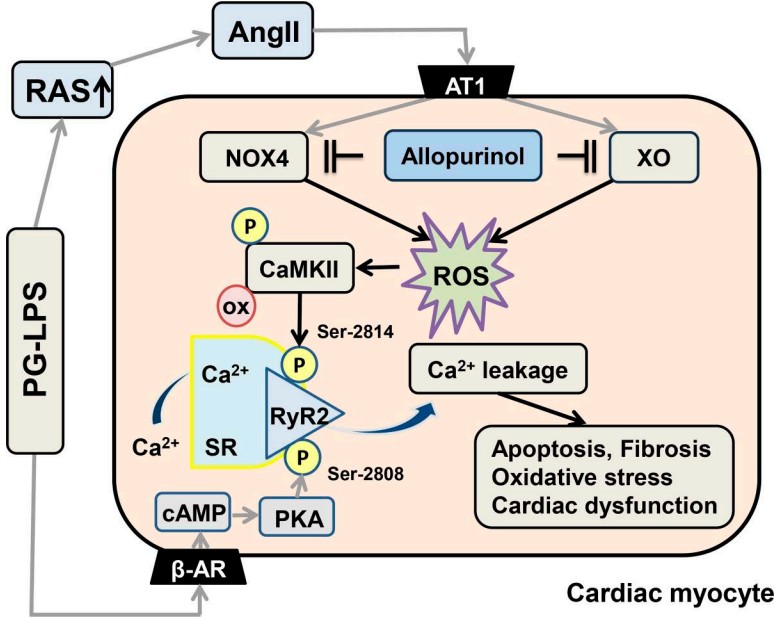

**Fig 7. Schematic illustration of the proposed role of XO and NOX4 in the heart of PG-LPS-treated mice.** PG-LPS induces expression of XO and NOX4, leading to ROS production, which mediates CaMKII activation and RyR2 phosphorylation (Ser-2814). We previously demonstrated that PG-LPS might induce myocardial ROS production and $Ca^{2+}$-mishandling via activation of the RAS [17] and cAMP/PKA signaling [64]. Our current study indicates that allopurinol might have a protective effect against PG-LPS-mediated cardiac dysfunction by blocking the increase of ROS generation by XO and NOX4 and $Ca^{2+}$ leakage via altered RyR2 phosphorylation in mice. Solid black lines represent findings in this study and solid gray lines represent findings reported previously [17,64]. β-AR, β-adrenergic receptor; SR, sarcoplasmic reticulum; RyR2, ryanodine receptor 2; AT1, angiotensin II type 1 receptor; PKA, protein kinase A; ROS, reactive oxygen species; cAMP, cyclic AMP.

by persistent subclinical exposure to PG-LPS might be caused by myocardial ROS production derived from XO and NOX4 via activation of the RAS (Fig 7).

In this study, we observed CaMKII-mediated RyR2 phosphorylation (Ser-2448) and protein kinase A-mediated RyR2 phosphorylation (Ser-2808) in the heart of PG-LPS-treated mice, which might be associated with SR $Ca^{2+}$ leakage via the RyR2 channel and heart failure [63]. We previously demonstrated that persistent subclinical exposure to PG-LPS in mice at the same dose used in this study might induce the cAMP/protein kinase A and cAMP/CaM-KII signaling pathways (Fig 7) [64], and these findings are consistent with the observation of phosphorylation at serine-2814 and serine-2808.

Overall, our results suggest that the XO inhibitor allopurinol might have a protective effect against periodontitis-mediated CVD by blocking the increase of ROS generation mediated by XO and NOX4 in PG-LPS-treated mice.

## Supporting information

**S1 Data. Representative full-length immunoblots shown in the main article.**
(PDF)

**S2 Data. Food/water consumption and negative/positive control sections of TUNEL staining and 8-OHdG immunostaining.**
(PDF)

**S3 Data. Normality test using the Shapiro-Wilk test.**
(XLSX)

## Author contributions

**Conceptualization:** Akinaka Morii, Ichiro Matsuo, Kenji Suita, Yoshiki Ohnuki, Satoshi Okumura.

**Formal analysis:** Akinaka Morii, Ichiro Matsuo, Yasuharu Amitani, Satoshi Okumura.

**Funding acquisition:** Ichiro Matsuo, Kenji Suita, Yoshiki Ohnuki, Aiko Ito, Megumi Nariyama, Satoshi Okumura.

**Investigation:** Akinaka Morii, Ichiro Matsuo, Kenji Suita, Go Miyamoto, Mariko Abe, Takao Mitsubayashi, Yoshio Hayakawa.

**Methodology:** Akinaka Morii, Ichiro Matsuo, Kenji Suita, Yoshiki Ohnuki, Misao Ishikawa, Yasumasa Mototani, Ren Matsubara.

**Supervision:** Kazuhiro Gomi, Takatoshi Nagano, Satoshi Okumura.

**Writing – original draft:** Satoshi Okumura.

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
