## [Decision Letter · Decision Letter 0]

10 Jul 2024

PONE-D-24-20166Allopurinol attenuates development of Porphyromonas gingivalis LPS-induced cardiomyopathy in micePLOS ONE

Dear Dr. Okumura,

Thank you for submitting your manuscript to PLOS ONE. After careful consideration, we feel that it has merit but does not fully meet PLOS ONE’s publication criteria as it currently stands. Therefore, we invite you to submit a revised version of the manuscript that addresses the points raised during the review process.

Your manuscript was reviewed by two experts and both of them  provided positive feed with minor comments.

We look forward to receiving your revised manuscript.

Kind regards,

Partha Mukhopadhyay, Ph.D.

Section Editor

PLOS ONE

Journal Requirements:

-https://doi.org/10.1038/s41598-023-43099-6

-https://doi.org/10.1371/journal.pone.0292624

In your revision ensure you cite all your sources (including your own works), and quote or rephrase any duplicated text outside the methods section. Further consideration is dependent on these concerns being addressed.

Reviewers' comments:

Reviewer's Responses to Questions

**Comments to the Author**

1. Is the manuscript technically sound, and do the data support the conclusions?

Reviewer #1: Partly

Reviewer #2: Yes

2. Has the statistical analysis been performed appropriately and rigorously? 

Reviewer #1: Yes

Reviewer #2: Yes

3. Have the authors made all data underlying the findings in their manuscript fully available?

Reviewer #1: Yes

Reviewer #2: Yes

4. Is the manuscript presented in an intelligible fashion and written in standard English?

Reviewer #1: Yes

Reviewer #2: Yes

5. Review Comments to the Author

Reviewer #1: In the manuscript, the authors utilize a PG-LPS-induced cardiac dysfunction model to investigate the role of oxidative stress in the pathogenesis of cardiovascular disease in periodontitis patients. The results indicate that treatment with the xanthine oxidase inhibitor allopurinol improves cardiac phenotype in PG-LPS-treated mouse. Detailed comments are listed below:

Major comments:

1. The results suggest that ROS is the key factor mediating the progression of heart disease in the PG-LPS model. To enhance reliability, the authors should provide additional evidence beyond 8-OHdG staining to support that allopurinol decreases ROS content in the heart. Suggested methods include qPCR for oxidative stress markers (e.g., gp22phox, gp40phox, gp47phox, gp91phox), dihydroethidium (DHE) staining, myocardial 3-nitrotyrosine (3-NT) content measurement, or immunohistochemistry of 3-NT.

2. Figure 6 indicates that calcium leakage is the trigger for heart dysfunction in PG-LPS-treated mice. Besides CaMKII and RyR2 phosphorylation, further evidence supporting calcium handling impairment in cardiomyocytes would be beneficial, such as Recan1.4 mRNA levels, which indicate increased cytosolic calcium.

3. Allopurinol acts as a competitive inhibitor of xanthine oxidase at low concentrations and as a noncompetitive inhibitor at high concentrations. According to the literature, allopurinol is expected to inhibit the activity of xanthine oxidase without affecting its expression level. However, the manuscript indicates that allopurinol treatment decreases XO expression, which requires explanation.

4. Although allopurinol is an XO inhibitor, Figures 5A and 6 suggest it also inhibits NOX4. The authors need to explain this observation.

5. Several antibody details provided in the manuscript are incorrect: catalogue number #AB765P is for Collagen type 1, not CaMKII; #NB600-549 is for PTTG1, not collagen 3; #AB767-P has no related antibody; #ab1092 is for RFXANK, not XO; and #2775 is for LC3B, not BAX.

Minor comments:

1. Please verify the scale bar size in Figure 2A; it appears inconsistent with the images and does not seem to be 100 μm.

2. Please verify the image sizes in Figure 3A; the LPS group appears to have a different magnification compared to other groups.

3. The protein mass of RyR2 is 565 kDa, but the western blot shows a band at 350 kDa. The pan RyR2 antibody should show a band at 565 kDa according to the instructions.

4. In Figure 6, "Ca2+ leakagel" should be corrected to "Ca2+ leakage."

Reviewer #2: The authors used a model of periodontitis cardiomyopathy with a low dose of a specific LPS from P. gengivalis to investigate the participation of xanthine oxidase (XO) in this process through allopurinol treatment. They showed, through functional and molecular methods that XO inhibition could prevent the development of cardiomyopathy, remodeling, and oxidative stress.

This paper is straightforward, clear, and elegantly conducted. Nevertheless, I have some points I would like to be addressed.

Overall, the text is clear, yet some constructions are a bit odd or unusual. There are some typos and repeated sentences that need to be checked (second paragraph of Material and Methods, you bring up ‘was directly in drinking water’ twice). You should also respect the order of the parenthesis. From the most external to the most internal, the order should be {[()]}. Nevertheless, I liked the transparency of displaying the results in the text as well.

Did you use any kind of normality test? I would recommend using a normality test. Although your data seems to be all in the Gaussian distribution, it is always good to submit it to a statistical test. Also, this is important for determining which test you should be using. You mention in the western blot segment of the material and methods that you excluded outliers. Did you use any kind of statistical test (e.g. Grubbs) to determine which would be the outlier samples that could be excluded? If so, please, describe it in your statistical analysis segment.

Nevertheless, the manuscript is good and shall be considered for publication after those corrections.

6. PLOS authors have the option to publish the peer review history of their article (what does this mean? ). If published, this will include your full peer review and any attached files.

**Do you want your identity to be public for this peer review?** For information about this choice, including consent withdrawal, please see our Privacy Policy .

Reviewer #1: No

Reviewer #2: No

---

## [Author Response · Author response to Decision Letter 1]

8 Nov 2024

Journal Requirements:

2. We noticed you have some minor occurrence of overlapping text with the following previous publication(s), which needs to be addressed.

-https://doi.org/10.1038/s41598-023-43099-6

-https://doi.org/10.1371/journal.pone.0292624

Response:

Thank you. There is indeed some overlapping text in the method section as you indicated. We have modified the relevant sentences in the revised manuscript.

Reviewer #1:

In the manuscript, the authors utilize a PG-LPS-induced cardiac dysfunction model to investigate the role of oxidative stress in the pathogenesis of cardiovascular disease in periodontitis patients. The results indicate that treatment with the xanthine oxidase inhibitor allopurinol improves cardiac phenotype in PG-LPS-treated mouse. Detailed comments are listed below.

Major comments:

1. The results suggest that ROS is the key factor mediating the progression of heart disease in the PG-LPS model. To enhance reliability, the authors should provide additional evidence beyond 8-OHdG staining to support that allopurinol decreases ROS content in the heart. Suggested methods include qPCR for oxidative stress markers (e.g. ,gp22phox, gp40phox, gp47phox, gp91phox), dihydroethdium (DHE) staining, myocardial 3-nitrotyrosine (3-NT) content measurement, or immunohistochemistry of 3-NT.

Response-1:

ROS production by Nox4 is dependent on the p22phox protein expression level [1, 2]. We thus examined the expression of p22phox protein in the heart among the four groups and found that p22phox expression was significantly increased in the PG-LPS-treated group [Control (n = 4) vs. PG-LPS (n = 5): 100 ± 45 vs. 183 ± 47%, P < 0.05 vs. Control; ANOVA/Tukey-Kramer]. This increase was suppressed by allopurinol [PG-LPS (n = 5) vs. PG-LPS + allopurinol (n = 6): 183 ± 47 vs. 111 ± 28%, P < 0.05 vs. PG-LPS; ANOVA/Tukey-Kramer].

These data suggest that PG-LPS-induced ROS production might enhance the activity of Nox4 by upregulating the expression of p22phox, leading to increased ROS production via Nox4, while allopurinol treatment might decrease Nox4-dependent ROS production through a decrease of p22phox expression.

We incorporated the above data in Fig 5B and the results section of the revised manuscript with new references (Page 21, line 13-Page 22, line 6).

p91phox and 3-NT are markers of oxidative stress and protein oxidation damage [3, 4]. We thus examined the expression levels of p91phox and 3-NT in the heart among the four groups and found that both were significantly increased in the PG-LPS-treated group [p91phox : Control (n = 6) vs. PG-LPS (n = 7): 100 ± 54 vs. 301 ± 82% P < 0.01 vs. Control; 3-NT: Control (n = 4) vs. PG-LPS (n = 4): 100 ± 8 vs. 170 ± 39%, P < 0.05 vs. Control; ANOVA/Tukey-Kramer each]. Further, these increases were suppressed by allopurinol [p91phox: PG-LPS (n = 7) vs. PG-LPS + allopurinol (n = 7): 301 ± 82 vs. 132 ± 36%, P < 0.01 vs. PG-LPS; 3-NT: PG-LPS (n = 4) vs. PG-LPS + allopurinol (n = 5): 170 ± 39 vs. 101 ± 26%, P < 0.05 vs. PG-LPS; ANOVA/Tukey-Kramer].

These data also support the view that allopurinol has a protective action against oxidative stress induced by PG-LPS.

We have incorporated the above data in Fig 5C-5E and the results section of the revised manuscript with new references (Page 22, lines 8-18).

2. Figure 6 indicates that calcium leakage is the trigger for heart dysfunction in PG-LPS-treated mice. Besides CaMKII and RyR2 phosphorylation, further evidence supporting calcium handling impairment in cardiomyocytes would be beneficial, such as Recan1.4 mRNA levels, which increased cytosolic calcium.

Response-2

The maintenance of calcium (Ca2+) homeostasis during muscle contraction is requisite for optimal contractile function, and altered Ca2+ homeostasis might induce hyperactivation of calcineurin-NFAT signaling [5]. We thus examined the activation level of calcineurin-NFAT signaling in terms of the phosphorylation level on serine 265 of NFATc3, which is a Ca2+-handling protein involved in Ca2+ homeostasis in cardiac muscle. We found that the phosphorylation level was significantly decreased in the PG-LPS-treated group [Control (n = 4) vs. PG-LPS (n = 4): 100 ± 26 vs. 40 ± 14%, P < 0.01 vs. Control; ANOVA/Tukey-Kramer]. Further, this decrease was suppressed by allopurinol [PG-LPS (n = 4) vs. PG-LPS + allopurinol (n = 4): 40 ± 14 vs. 94 ± 28%, P < 0.05 vs. PG-LPS; ANOVA/Tukey-Kramer] (Fig 6E).

These data indicated that allopurinol might protect the heart from Ca2+ handling impairment induced by PG-LPS.

We incorporated the above data in Fig 6E and the results section of the revised manuscript (Page 25, lines 1-12).

3. Allopurinol acts as a competitive inhibitor of xanthine oxidase at low concentrations and as a noncompetitive inhibitor at high concentrations. According to the literature, allopurinol is expected to inhibit the activity of xanthine oxidase without affecting its expression level. However, the manuscript indicates that allopurinol treatment decreases XO expression level, which requires explanation.

Response-3

The effect of allopurinol on XO expression level is controversial: some studies have found that allopurinol inhibits XO activity without reducing its expression level [6, 7], while others have found a reduction of its expression level [8, 9].

We incorporated the above sentences in the discussion section of the revised manuscript with new references (Page 27, lines 5-7).

4. Although allopurinol is a XO inhibitor, Figure 5A and 6 suggest it also inhibits NOX4. The authors need to explain this observation.

Response-4

It has been shown that allopurinol treatment reduces oxidative stress by decreasing the expression level of Nox4, in addition to XO, in a mouse model of Marfan syndrome [10].

We incorporated the above sentences in the results section of the revised manuscript (Page 20, lines 16-18).

5. Several antibody details provided in the manuscript are incorrect: catalog number #AB765P is for Collagen type 1, not CaMKII; NB600-549 is for PTTG1, not collagen 3; #AB767-P has no related antibody; #ab1092 is for RFXANK, not XO; and #2775 is for LC3B, not BAX.

Response-5

Thank you. We rechecked the catalogue numbers and sources, and made the following corrections.

1) oxidized calmodulin kinase II (Met-2812/282) AB765P → 071387 Millipore→Merck (San Jose, CA, USA)

2) collagen type 3 NB600-549 → NB600-594

3) collagen I AB767-P →AB765P Millipore → Merck (San Jose, CA, USA)

4) XO ab1092 → ab109235

5) BAX #2775 → #2772

Minor comments:

1. Please verify the scale bar size in Figure 2A; it appears inconsistent with the images and does not seem to be 100 μm.

Response-1

Thank you. We re-checked it and modified the scale bar in Fig 2A in the revised manuscript.

2. Please verify the images size in Figure 3A; the LPS group appears to have a different magnification compared to the other groups.

Response-2

Thank you. We replaced the photo of the LPS group in Fig 3A.

3. The protein mass of RyR2 is 565 kDa, but the western blot shows a band at 350 kDa. The pan RyR2 antibody should shows band at 565 kDa according to the instructions.

Response-3:

We performed a western blot of RyR2 using high- and low- molecular-weight markers (as is our usual practice) and confirmed that the protein mass of RyR2 is 565kDa.

We thus modified the molecular weight of RyR2 (350kDa → 565kDa) in Fig 6C and 6D of the revised manuscript. We apologize for the error.

4. In Figure 6 ”Ca2+ leakagel” should be corrected to “Ca2+ leakage”.

Response-4

Thank you. We corrected it (Fig 7 of the revised manuscript).

Reviewer #2:

The authors used a model of periodontitis cardiomyopathy with a low dose of a specific LPS from P. gengivalis to investigate the participation of xanthine oxidase (XO) in this process through allopurinol treatment. They showed, through functional and molecular methods that XO inhibition could prevent the development of cardiomyopathy, remodeling, and oxidative stress. This paper is straightforward, clear, and elegantly conducted. Nevertheless, I have some points I would like to be addressed.

１．Overall, the text is clear, yet some constructions are a bit odd or unusual. There are some typos and repeated sentences that need to be checked (second paragraph of Materials and Methods, you bring up ‘was directly in drinking water’ twice).You should also respect the order of the parenthesis. From the most external to the most internal, the order should be {[()]}. Nevertheless, I liked the transparency of displaying the results in the text as well.

Response-1:

We have carefully checked for typos, repeated phases and the order of parentheses as the reviewer suggested, and corrected the following points.

1) once daily for 1 1week →once daily for 1 week (Page 7, lines 9-10)

2) was directly dissolved in drinking water was directly dissolved in drinking water--- →was directly dissolved in drinking water (Page 7, lines 13-14)

3) pH7.5 → pH 7.5 (Page 9, line 2)

4) incubated with H2O2 →incubated with 0.3% H2O2 (Page 11, line 1)

5) phospho-RyR2 (1:5000) (Ser-2814, #A010-31) →phospho-ryanodine receptor 2 (RyR2) (1:5000) (Ser-2814, #A010-31) (Page 12, lines 13-14)

6) tibial length ration →tibial length ratio (Page 15, line 9)

7) 2.5 ± 0.4→2.5 ± 0.4% (Page 16, line 15)

8) 0.9 ± 0.2→0.9 ± 0.2% (Page 16, line 18)

9) ryanodine receptor 2 (RyR2) →RyR2 (Page 24, line 1)

10) renin-angiotensin system→renin-angiotensin system (RAS) (Page 27, Line 15)

11) renin-angiotensin system (RAS) →RAS (Page 28, line 4)

2-(1). Did you use any kind of normality test? I would recommend using a normality test. Although your data seems to be all in the Gaussian distribution, it is always good to submit it to a statistical test. Also, this is important for determining which test you should be using.

Response-2-(1):

We asked Dr. Yasuharu Amitani, a statistician, who is included as a co-author in the revised manuscript, to examine the validity of our statistical treatment in response to this comments. We performed the Shapiro-Wilk test to evaluate if the sample shows a normal distribution [11]. When the test showed that the distribution was not normal, we used a non-parametric test for analysis.

We incorporated the following sentences in the method section of the revised manuscript (Page 13, line 10-Page 14, line 2).

Statistical analysis

The Shapiro-Wilk test was performed to evaluate if the sample showed a normal distribution (S3 Data) [11]. When the distribution was not normal, we used a non-parametric test for analysis (Steel-Dwass test) [Fig 1C, Fig 2C and Table 1 (LVPWTd)]. Comparisons were performed using one-way ANOVA followed by the Tukey-Kramer post hoc test (hereafter abbreviated as ANOVA/Tukey-Kramer) (Fig 1B, 1D, 1E, Fig 2B-2D, Fig 3C, Fig 4B, Fig 5, Table 1 and S1 Fig of S2 Data) or non-parametric one-way ANOVA followed by the Steel-Dwass post hoc test (hereafter abbreviated as nonparametric ANOVA/Steel-Dwass) [Fig 1C, Fig 2C and Table 1 (LVPWTd)] [12]. Differences were considered significant when P < 0.05.

2-(2): You mention in the western blot segment of the material and methods that you excluded outliners. Did you use any kind of statistical test analysis (e.g. Grubbs) to determine which would be the outlier samples that could be excluded? If so, please, describe it in your statistical test analysis segment.

Response-2-(2):

We performed the Smirnov-Grubbs test to excluded outliers (extremely low or high values, compared to others in the same groups) [13].

We thus incorporated the following sentences in the method section of the revised manuscript with a new reference (Page 13, lines 7-8).

---- we excluded outliers (extremely low or high values, compared to others in the same groups) using the Smirnov-Grubbs test [13].

Nevertheless, the manuscript is good and shall be considered for publication after those corrections.

References

1. Martyn KD, Frederick LM, von Loehneysen K, Dinauer MC, Knaus UG. Functional analysis of Nox4 reveals unique characteristics compared to other NADPH oxidases. Cell Signal. 2006;18(1):69-82. https://doi:10.1016/j.cellsig.2005.03.023. PMID: 15927447.

2. Fukui T, Ishizaka N, Rajagopalan S, Laursen JB, Capers Qt, Taylor WR, et al. p22phox mRNA expression and NADPH oxidase activity are increased in aortas from hypertensive rats. Circ Res. 1997;80(1):45-51. https://doi:10.1161/01.res.80.1.45. PMID: 8978321.

3. Bedard K, Krause KH. The NOX family of ROS-generating NADPH oxidases: physiology and pathophysiology. Physiol Rev. 2007;87(1):245-313. https://doi:10.1152/physrev.00044.2005. PMID: 17237347.

4. Bandookwala M, Thakkar D, Sengupta P. Advancements in the Analytical quantification of nitroxidative stress biomarker 3-nitrotyrosine in biological matrices. Crit Rev Anal Chem. 2020;50(3):265-89. https://doi:10.1080/10408347.2019.1623010. PMID: 31177807.

5. Ravel-Chapuis A, Belanger G, Cote J, Michel RN, Jasmin BJ. Misregulation of calcium-handling proteins promotes hyperactivation of calcineurin-NFAT signaling in skeletal muscle of DM1 mice. Hum Mol Genetics. 2017;26(12):2192-206. https://doi:10.1093/hmg/ddx109. PMID: 28369518.

6. Xiao J, She Q, Wang Y, Luo K, Yin Y, Hu R, et al. Effect of allopurinol on cardiomyocyte apoptosis in rats after myocardial infarction. Eur J Heart Fail. 2009;11(1):20-7. Epub 2009/01/17. https://doi:10.1093/eurjhf/hfn003. PMID: 19147453.

7. Rajesh M, Mukhopadhyay P, Bátkai S, Mukhopadhyay B, Patel V, Haskó G, et al. Xanthine oxidase inhibitor allopurinol attenuates the development of diabetic cardiomyopathy. J Cell Mol Med. 2009;13(8b):2330-41. https://doi:10.1111/j.1582-4934.2008.00564.x. PMID: 19175688.

8. Wang Z, Ding J, Luo X, Zhang S, Yang G, Zhu Q, et al. Effect of allopurinol on myocardial energy metabolism in chronic heart failure rats after myocardial infarct. Int Heart J. 2016;57(6):753-9. https://doi:10.1536/ihj.16-149. PMID: 27818481.

9. Yang Y, Zhao J, Qiu J, Li J, Liang X, Zhang Z, et al. Xanthine oxidase inhibitor allopurinol prevents oxidative stress-mediated atrial remodeling in alloxan-induced diabetes mellitus rabbits. J Am Heart Assoc. 2018;7(10). https://doi:10.1161/jaha.118.008807. PMID: 29720500.

10. Rodríguez-Rovira I, Arce C, De Rycke K, Pérez B, Carretero A, Arbonés M, et al. Allopurinol blocks aortic aneurysm in a mouse model of Marfan syndrome via reducing aortic oxidative stress. Free Radic Biol Med. 2022;193(Pt 2):538-50. https://doi:10.1016/j.freeradbiomed.2022.11.001. PMID: 36347404.

11. Shapiro SW, MB. An analysis of variance test for normality (Complete samples). Biometrika. 1965;52(3/4):591-611.

12. Midway S, Robertson M, Flinn S, Kaller M. Comparing multiple comparisons: practical guidance for choosing the best multiple comparisons test. PeerJ. 2020;8:e10387. https://doi:10.7717/peerj.10387. PMID: 33335808.

13. Grubbs F. Sample criteria for testing outlying observations. Ann Math Statist. 1950;21:27-58.

---

## [Decision Letter · Decision Letter 1]

17 Dec 2024

PONE-D-24-20166R1Allopurinol attenuates development of Porphyromonas gingivalis LPS-induced cardiomyopathy in micePLOS ONE

Dear Dr. Okumura,

Thank you for submitting your manuscript to PLOS ONE. After careful consideration, we feel that it has merit but does not fully meet PLOS ONE’s publication criteria as it currently stands. Therefore, we invite you to submit a revised version of the manuscript that addresses the points raised during the review process.

Your manuscript was reviewed by same reviewers and one of  them suggested  few minor modifications. Please address those comments and a quick editorial decision without further review will taken after addressing those comments.

We look forward to receiving your revised manuscript.

Kind regards,

Partha Mukhopadhyay, Ph.D.

Section Editor

PLOS ONE

Journal Requirements:

Reviewers' comments:

Reviewer's Responses to Questions

**Comments to the Author**

1. If the authors have adequately addressed your comments raised in a previous round of review and you feel that this manuscript is now acceptable for publication, you may indicate that here to bypass the “Comments to the Author” section, enter your conflict of interest statement in the “Confidential to Editor” section, and submit your "Accept" recommendation.

Reviewer #1: All comments have been addressed

Reviewer #2: All comments have been addressed

2. Is the manuscript technically sound, and do the data support the conclusions?

Reviewer #1: Yes

Reviewer #2: Yes

3. Has the statistical analysis been performed appropriately and rigorously? 

Reviewer #1: Yes

Reviewer #2: Yes

4. Have the authors made all data underlying the findings in their manuscript fully available?

Reviewer #1: Yes

Reviewer #2: Yes

5. Is the manuscript presented in an intelligible fashion and written in standard English?

Reviewer #1: Yes

Reviewer #2: Yes

6. Review Comments to the Author

Reviewer #1: The authors have already revised the manuscript which meets my previous requirements. But there are still some minor problems as outlined below.

1.The figure title of Fig5 and Fig6 are the same.

2. In Fig7, there’s an arrow from allopurinol to ROS, and it appears that allopurinol increases ROS production which contradicts the findings. The authors need to modify the schematic diagram to make it more intuitive for the readers.

3. In the legend of Fig7, the author focused on describing their previous discovery without mentioning their finding that allopurinol can protect heart from PG-LPS induced cardiac dysfunction.

Reviewer #2: The authors properly acknowledged the concerns raised in my review. They improved considerably the quality of the text with the figures. I am glad to recommend the acceptance of this manuscript for publication.

7. PLOS authors have the option to publish the peer review history of their article (what does this mean? ). If published, this will include your full peer review and any attached files.

**Do you want your identity to be public for this peer review?** For information about this choice, including consent withdrawal, please see our Privacy Policy .

Reviewer #1: No

Reviewer #2: **Yes: ** Bruno Paes Leme Ferreira

---

## [Author Response · Author response to Decision Letter 2]

23 Dec 2024

Reviewer #1:

The authors have already revised the manuscript which meets my previous requirements. But there are still some minor problems as outlined below.

1. The figure title of Fig5 and Fig6 are the same.

Response-1: Thank you for pointing this out. We modified the title of Fig 6 as shown below.

Effects of allopurinol on PG-LPS-induced phospho-CaMKII, ox-CaMKII, phospho-RyR2 (Ser-2814), phospho-RyR2 (Ser-2808) and phospho-NFATc3 in cardiac muscle.

2. In Fig7, there’s an arrow from allopurinol to ROS, and it appears that allopurinol increases ROS production which contradicts the findings. The authors need to modify the schematic diagram to make it more intuitive for the readers.

Response-2: We deleted the line from allopurinol to ROS in Fig 7. Instead, we drew new lines from NOX4 to ROS and XO to ROS, which should clarify the situation for readers.

3. In the legend of Fig7, the authors focused on describing their previous discovery without mentioning their finding that allopurinol can protect heart from PG-LPS induced cardiac dysfunction.

Response-3: We modified the figure legend as shown below to include this information.

Schematic illustration of the proposed role of XO and NOX4 in the heart of PG-LPS-treated mice.

PG-LPS induces expression of XO and NOX4, leading to ROS production, which mediates CaMKII activation and RyR2 phosphorylation (Ser-2814). We previously demonstrated that PG-LPS might induce myocardial ROS production and Ca2+-mishandling via activation of the RAS [1] and cAMP/PKA signaling [2]. Our current study indicates that allopurinol might have a protective effect against PG-LPS-mediated cardiac dysfunction by blocking the increase of ROS generation by XO and NOX4 and Ca2+ leakage via altered RyR2 phosphorylation in mice. Solid black lines represent findings in this study and solid gray lines represent findings reported previously [1, 2]. RAS; renin-angiotensin-aldosterone system; β-AR, β-adrenergic receptor; SR, sarcoplasmic reticulum; RyR2, ryanodine receptor 2; AT1, angiotensin II type 1 receptor; PKA, protein kinase A; ROS, reactive oxygen species; cAMP, cyclic AMP

Reviewer #2:

The authors properly acknowledged the concerns raised in my review. They improved considerably the quality of the text with the figures. I am glad to recommend the acceptance of this manuscript for publication.

Response: Thank you.

References

1. Kiyomoto K, Matsuo I, Suita K, Ohnuki Y, Ishikawa M, Ito A, et al. Oral angiotensin-converting enzyme inhibitor captopril protects the heart from Porphyromonas gingivalis LPS-induced cardiac dysfunction in mice. PloS one. 2023;18(11):e0292624. https://doi:10.1371/journal.pone.0292624. PMID: 37983238.

2. Matsuo I, Ohnuki Y, Suita K, Ishikawa M, Mototani Y, Ito A, et al. Effects of chronic Porphylomonas gingivalis lipopolysaccharide infusion on cardiac dysfunction in mice. J Oral Biosci. 2021;63(4):394-400. https://doi:10.1016/j.job.2021.10.001. PMID: 34757204.

---

## [Editor Report · Decision Letter 2]

9 Jan 2025

Allopurinol attenuates development of Porphyromonas gingivalis LPS-induced cardiomyopathy in mice

PONE-D-24-20166R2

Dear Dr. Okumura,

We’re pleased to inform you that your manuscript has been judged scientifically suitable for publication and will be formally accepted for publication once it meets all outstanding technical requirements.

Kind regards,

Partha Mukhopadhyay, Ph.D.

Section Editor

PLOS ONE
---

## [Editor Report · Acceptance letter]

PONE-D-24-20166R2

PLOS ONE

Dear Dr. Okumura,

I'm pleased to inform you that your manuscript has been deemed suitable for publication in PLOS ONE. Congratulations! Your manuscript is now being handed over to our production team.

Kind regards,

on behalf of

Dr. Partha Mukhopadhyay

Section Editor

PLOS ONE